# Papaya Leaf Curl Virus (PaLCuV) Infection on Papaya (*Carica papaya* L.) Plants Alters Anatomical and Physiological Properties and Reduces Bioactive Components

**DOI:** 10.3390/plants11050579

**Published:** 2022-02-22

**Authors:** Sumit K. Soni, Manoj Kumar Mishra, Maneesh Mishra, Swati Kumari, Sangeeta Saxena, Virendra Shukla, Sudeep Tiwari, Pramod Shirke

**Affiliations:** 1Crop Improvement and Biotechnology Division, ICAR-Central Institute for Subtropical Horticulture, Rehmankhera, P.O. Kakori, Lucknow 226101, India; sumit.soni@icar.gov.in (S.K.S.); swati19111987@gmail.com (S.K.); 2Plant Physiology Laboratory, CSIR-National Botanical Research Institute, Rana Pratap Marg, Lucknow 226001, India; manoj30biotech@gmail.com (M.K.M.); pashirke@nbri.res.in (P.S.); 3Department of Biotechnology, Babasaheb Bhimrao Ambedkar University, Vidya Vihar, Rae Bareli Road, Lucknow 226025, India; dr_sangeeta_saxena@yahoo.com (S.S.); shuklavirendra121@gmail.com (V.S.); 4Department of Biochemistry and Molecular Biology, Faculty of Medicine, IMRIC, The Hebrew University of Jerusalem, P.O. Box 12271, Jerusalem 91120, Israel; 5Department of Geography and Environmental Development, Ben Gurion University of the Negev, P.O. Box 653, Beer-Sheva 84105, Israel; sudeep@post.bgu.ac.il

**Keywords:** *Carica papaya* L., PaLCuV, anatomy, physiology, bioactive properties

## Abstract

Papaya leaves are used frequently for curing scores of ailments. The medicinal properties of papaya leaves are due to presence of certain bioactive/pharmacological compounds. However, the papaya leaf curl virus (PaLCuV), a geminivirus, is a major threat to papaya cultivation globally. During the present investigation, we observed that PaLCuV infection significantly altered the anatomy, physiology, and bioactive properties of papaya leaves. As compared to healthy leaves, the PaLCuV-infected leaves were found to have reduced stomatal density (76.83%), stomatal conductance (78.34%), photosynthesis rate (74.87%), water use efficiency (82.51%), chlorophyll (72.88%), carotenoid (46.63%), osmolality (48.55%), and soluble sugars (70.37%). We also found lower enzymatic activity (superoxide dismutase (SOD), ascorbate peroxidase (APX), and catalase (CAT)—56.88%, 85.27%, and 74.49%, respectively). It was found that the size of guard cells (50%), transpiration rate (45.05%), intercellular CO_2_ concentration (47.81%), anthocyanin (27.47%), proline content (74.17%), malondialdehyde (MDA) (106.65%), and electrolyte leakage (75.38%) was elevated in PaLCuV-infected leaves. The chlorophyll fluorescence analysis showed that the infected plant leaves had a significantly lower value of maximal quantum yield of photosystem II (PSII (Fv/Fm), photochemical quantum yield of photosystem I (PSI (Y(I)), and effective quantum yield of PSII (Y(II)). However, in non-photochemical quenching mechanisms, the proportion of energy dissipated in heat form (Y(NPQ)) was found to be significantly higher. We also tested the bioactivity of infected and healthy papaya leaf extracts on a *Caenorhabditis elegans* (*C. elegans*) model system. It was found that the crude extract of papaya leaves significantly enhanced the life span of *C. elegans* (29.7%) in comparison to virus-infected leaves (18.4%) on application of 100 µg/mL dose of the crude extract. Our research indicates that the PaLCuV-infected leaves not only had anatomical and physiological losses, but that pharmacological potential was also significantly decreased.

## 1. Introduction

Papaya (*Carica papaya* L.), belongs to the Caricaceae family and is an economically important crop, one that is known for its nutritional and medicinal properties [1]. People have used the papaya leaves for various medicinal purposes for hundreds of years. Traditionally, papaya leaves were used for the management of thrombocytopenia during dengue infection [2,3,4], and decoction is used for healing capabilities against cancer [5]. Recently, through the unlocking of their therapeutic potential by researchers, the demand for papaya has continuously increased in the pharmaceutical and wellness industries. The other proven therapeutic properties of papaya leaf include antimicrobial [6], anti-malarial [7], anti-tumor [8], anti-inflammatory [9], antioxidant [10], and immune modulator properties [11]; reducing blood sugar [12]; wound healing and gastro-protective effects [13]; and curing irregular menstruation, fever [14], colica, asthma, jaundice, and beriberi [15]. According to a World Health Organization (WHO)estimation, as a primary form of healthcare, approximately 80% of people from developing countries still keep their faith in the traditional/herbal medicinal system [16]. During the COVID-19 pandemic, the usages of herbal products have increased manifold. On the Indian subcontinent, the present pandemic scenario has forced people to rely on the use of medicinal plants directly in their day-to-day life. The remedial value of papaya leaf is due to the richness of various phytochemical compounds such as phenols, alkaloids, steroids, flavonoids, tannins, and quinines [17,18]. The leaves also have a number of biologically active compounds, i.e., papain, caricain, chymopapain, and glycine endopeptidase [19]. Moreover, it is also a rich source of proteins, lipids, carbohydrates, and vitamins. Owing to this reason, it can also be used as a nutritional supplement. In last few decades, due to the surge of dengue infection in South East Asia, papaya extract has emerged as one of most sought after plant-based products for the management of dengue, boosting platelet count in infected patients [19]. However, the growth of papaya leaves is adversely impacted by the Geminiviruses, known as papaya leaf curl virus (PaLCuV). The PaLCuV are transmitted to plants through whitefly (*Bemisia tabaci*) vector. In India, the disease was first noticed by Thomas and Krishnaswamy [20], whereas the mystery of responsible causal organism for disease was unlocked by Saxena et al. [21,22]. The diseases of PaLCuV-infected plants are characterized by wrinkled and curled leaves that roll downward or inward and appear as an inverted cup. The leaf becomes leathery, rigid, and reduced in size with thickened vein and zig-zag twisted petioles. Moreover, infected plants show defoliation, fail to bear flowers or fruits, and have restricted growth during the advanced stages of infection [23].

India is the leading papaya-producing country in the world. Infection with PaLCuV can severely affect its commercial production, and moreover, it can also impact the pharmaceutical industries. Thus, infection with PaLCuV causes greater loss of crops, which may be a bottleneck for the desired demand of industries. The alteration in physiology and bioactive properties of papaya leaves are due to PaLCuV infection; however, the impact and detailed information is still illusive. Therefore, it is necessary to study the impact of infection on physiological and biochemical attributes that will help in future in order to devise a suitable management strategy. Considering the medicinal and industrial value of papaya leaf, the present study (i) aimed to unravel the effect of PaLCuV infection on physiological parameters, and (ii) also assumed that the infected leaf has altered bioactive properties. In order to assess bio-potentials of healthy and infected leaves, we included a simple eukaryotic model organism, *Caenorhabditis elegans*, in the study.

## 2. Results and Discussion

### 2.1. Morphological Characterization and PCR Analysis of Symptomatic Leaves

The symptomatic plants were characterized by wrinkled and twisted leaves with the presence of yellow veins (Figure 1B). The petioles were short in size and also showed a thick and twisted shape. The plants either had no fruits or had smallfruits with distorted shape.

To confirm the PaLCuV infection in symptomatic papaya plants, we isolated the DNA from leaves of both healthy (control) and symptomatic plants, which was followed by amplification of virus gene using PaLCuV gene-specific primers using PCR. The 1.6 kb amplicon with PaLCuV gene were amplified from the isolated DNA samples of symptomatic plant leaves while no amplification of genes was observed in the DNA samples of healthy plant leaves (Figure 2). This result clearly indicates that the aforementioned symptoms in plants occurred only due to infection of PaLCuV.

### 2.2. Stomata Density and Guard Cell Size

The stomatal density and guard cell size were measured in both healthy and symptomatic leaves using a light microscope. The microscopic observation revealed that the symptomatic (infected) plants showed drastic reduction of stomata numbers (<250/mm^2^). As compared to healthy ones (≈1000/mm^2^), they were reduced by around three-quarters (76.83%) (Figure 3A). Unlike the stomatal density, the length of guard cells in symptomatic (infected) leaves was also increased by approximately one-half (29–30 µm) when compared with healthy leaves of 19–20 µm (Figure 3B). In healthy papaya leaf cells, the concentration of starch grains was higher than the infected leaf cells. The concentration of starch is responsible for maintaining the proper shape of the guard cells and their function, i.e., opening and closing of the stomatal aperture via shifting of turgor pressure. The changes in starch concentration causes alteration in guard cells shape and size, and consequently wrinkling of leaves occur, which further strengthens our results and might be a probable reason for the reduction of stomata [24].

### 2.3. Photosynthesis, Transpiration, Stomatal Conductance, and Intercellular CO_2_ Concentration in PaLCuV-Infected Leaves

An inverse relationship was observed between virus infection and photosynthesis rate. A significant reduction (74.87%) in photosynthesis rate was observed in PaLCuV-infected leaves (Figure 4A) as compared to healthy ones. The changes in photosynthesis rate may have been due to damage and irregularity of photosynthetic pigments caused by DNA virus-encoded protein or blockage at any point of the photosynthetic energy cascade by inhibition of the downstream enzyme (s) [25,26]. As a result of alteration in photosynthetic pigment or enzyme involved in the photosynthesis process, the chlorosis and yellowing of leaves occurred in infected leaves. Previous reports on virus-infected leaves were also on par with our results, wherein researchers reported that several physiological changes such as stomatal closure were lower gas exchange are associated [27], further adding to the reduction of photosynthesis rate. The wrinkling and curling of leaves resulted in reduced leaf surface area in terms of light exposure and consequently also decreased the rate of photosynthesis.

A noticeable reduction (78.34%) in stomatal conductance was observed in severely infected leaves (Figure 4C). The decrease in stomatal conductance could have been due to alteration of stomatal morphology (stomatal density, size of guard cells, and stomatal pores) and development. The changes in stomatal morphology and development in other virus-infected plants have been previously reported [28]. The possible molecular link between stomatal development and plant virus infections was established by Ruggenthaler et al. [29]. Zhu et al. [30] reported that presence of AM fungi improved the stomatal conductance and stomatal density. Thus, possibilities of changes of stomatal conductance and stomatal density might be due to depletion of micro-flora inside the plant cells, which cannot be ignored. The virus might have affected the endophytic population and consequently indirectly altered the stomatal conductance activity.

The transpiration rate (E) and intercellular CO_2_ concentration (Ci) of infected leaves were also altered, and they were found to be higher (45.05% and 47.81%, respectively) in infected leaves (Figure 4B,D).

In the present study, microscopic observation showed that the healthy plant leaves demonstrated maximum stomatal density with fine opening at the lower surface, which was better able to regulate CO_2_ uptake for photosynthetic carbon assimilation and water loss through transpiration. Aside from this, PaLCuV-infected leaves had comparatively less stomatal density with a large guard cell size. However, PaLCuV infection in papaya caused the distortion in the shape of the leaves. Hence, the anatomy of leaf cell was wrinkled and the arrangement of the surrounding epithelial cell was disturbed, resulting in open guard cells. This pattern of arrangement of cells may have altered the size and turgidity of guard cells in PaLCuV-symptomatic leaves.

Overall, infected plants have abnormal anatomy and physiology, including a compromised photosynthetic apparatus.

### 2.4. Water Use Efficiency in PaLCuV-Affected Leaves

Water use efficiency (WUE) is the ratio of CO_2_ assimilation rate to transpiration rate at the leaf level. It was revealed during the investigation that leaf curl virus infection drastically reduced (82.51%) the water use in infected plants (Figure 5). Murray et al. [28] reported that plant viruses such as Tobacco mosaic virus (TMV) and Turnip vein-clearing virus (TVCV) infection influence stomatal development. The reduction in number of stomata in *Beta vulgaris* L. leaves infection of beet yellow virus (BYV) was also reported [31]. A probable connection of plant virus infections and stomatal development was reported by Ruggenthaler et al. [29]. The diminution in WUE may have been due to reduction of stomatal numbers in infected plants, which regulate the photosynthesis and transpiration rate, consequently modulating the WUE of plants.

### 2.5. Effect on Chlorophyll, Anthocyanin, and Carotenoids in PaLCuV-Infected Leaves

The content of chlorophyll was decreased progressively with the increased infection and symptom development in infected plants. Total chlorophyll content in infected plant was found to be lower (72.88%) as compared to healthy (control) plants (Figure 6A).

Our results are in accordance with observations reported earlier [32,33]. Liu et al. [32] established the role played by plant virus in chlorophyll degradation via upregulation of chlorophyll degradation transcript genes. Thus, the reduction in chlorophyll content in infected papaya plants might have been due to the interference of the virus in the plant’s molecular machinery.

PaLCuV infection also altered the other plants’ pigment content such as carotenoids and anthocyanin. As compared to healthy plants, a significant reduction (46.63%) in carotenoid content was observed in severely infected plants (Figure 6C). The lowering of carotenoid content/reduction in carotenoid production in virus-infected plants was also reported earlier [34]. Ibdah et al. [34] further reported that plant virus downregulated the enzyme phytoene desaturase, a key enzyme involved in carotenoid biosynthesis.

The anthocyanin content was found to be higher (27.47%) in PaLCuV-infected plants rather than healthy (control) plants (Figure 6D). These results are in accordance with those previously reported on infected grapevine leaves [35]. The authors found that virus infection on plants modulated the flavonoid biosynthesis pathway and consequently increased the anthocyanin content [35]. The higher anthocyanin content in our results might have been due to upregulation of genes involved in anthocyanin biosynthesis.

### 2.6. Comparison of Chlorophyll Fluorescence in Infected Plant and Control Plant Leaves

Chlorophyll fluorescence analysis showed that infected plant leaves showed significantly lower Fv/Fm (Y(I) and Y(II)). Conversely, Y(NPQ) was notably higher in infected plant leaves. The lower Fv/Fm and Y(II) in infected leaves indicated potential photo damage, but increased Y(NPQ) was apparently sufficient in preventing irreversible PSII center damage. Reduced Y(II) and increased NPQ adjustment with excess excitation dissipation was through zeaxanthin feedback mechanisms. A high NPQ can compensate for decrease of Y(II) and can even cause a lowering of Quantum yield of non-regulated energy dissipation in PSI (Y(NO)). Quantum yield of non-photochemical energy dissipation due to donor side limitation (Y(ND)) was observed as being significantly higher in infected plant leaves. Quantum yield of non-photochemical energy dissipation due to acceptor side limitation (Y(NA)) did not show significant differences between control and infected plants (Figure 7).

The fluorescence property of chlorophyll is frequently used in examining the photosynthetic machinery and sightedness of the photosynthesis network under both abiotic and biotic stresses [36].

Previously, it was considered that the value of Fv/Fm was significantly lower in virus-infected *Oncidium* and *Solanum tuberosum* [37,38]. However, no alteration was observed in virus-infected *Eupatorium makinoi* plants [39]. The change in fm/fv value under biotic stress (virus infection) strongly supported our observations. As compared to healthy plants, the photosynthesis reaction, i.e., both dark and light reactions, were highly impaired in virus-infected plants. The progressive reduction (on the comparison of control with newly (infected 1 week ago) and old (infected 2 week ago) plants) of performance indices (PI) of photochemical Y(NPQ) reactions were observed (Figure 8). This result indicates that the reaction centers of photosynthesis apparatus were photochemically inactive due to damage caused by virus infection. As a consequence of this, the reduction in e^-^ transport capacity in PSII led to increased generation of ROS. Similar findings were reported by Rahoutei et al. [40]. The ROS generated in this way oxidizes plastid protein, especially protein engaged in PSII, such as D1, consequently inhibiting the photosynthesis process [41].

### 2.7. Proline Content and MDA Analysis

The PaLCuV-infected plant leaves had increased proline content relative to leaves from healthy plants (Figure 9A). Leaves of virus-infected plants showed proline concentration augmentation of 74.17% as compared to the control. Proline accumulation is a common metabolic response to both abiotic and biotic stresses, and when higher plants are exposed to stress, many plants accumulate high amounts of proline in tissues [42]. For instance, pathogen infection [43], salinity [44], high and low temperatures [45], and drought [46] may cause activation in numerous compounds in the cell or the proline production [47,48]. In our study, a higher amount of proline in infected plant was observed in comparison with the healthy plant. The results are supported by the finding of Chatterjee and Ghosh [49]. When plants are exposed to microbial pathogens, they produce reactive oxygen species (ROS) that induce programmed cell death in the plant cells surrounding the infection site to effectively wall off the pathogen and terminate the disease process. The amino acid proline may act as a potent scavenger of ROS, and this property of proline might prevent the induction of programmed cell death by ROS [50].

A further experiment was included in order to better understand the oxidative damage through lipid peroxidation analysis. The difference between oxidative damage of virus-infected plant and healthy plant was measured by MDA analysis. In the present work, we found virus-infected plants accumulated more MDA (106.65%) than healthy plants (Figure 9B). These data are consistent with previous report, wherein the higher MDA levels measured in Pepper mild mottle virus (PMMoV-S) infected plants could have been due to a higher accumulation of O^2–^ associated with the decreased SOD activities in different cell compartments [51]. The remarkable rise in oxidation of lipids, disturbance of biosynthesis of chloroplast, and inhibition of PSII are consistent with an accelerated senescence in PaLCuV-infected papaya plants.

### 2.8. Relative Electrical Conductivity and Osmotic Potential

Relative electrical conductivity indicates the relative percentage of cell membrane damage/leakage. Membrane damage/leakage could indirectly be evaluated by measuring solute leakage (electrolyte leakage) from cells [52] and membrane stability index [53]. The results clearly showed that infection with PaLCuV caused increased electrolyte leakage (75.38%) and decreased membrane stability index and osmolality (48.55%) of infected plants as compared with that of healthy plants (Figure 9C,D, respectively). The osmotic potential was also decreased in virus-infected leaves, which means the relative water content was decreased in virus-infected plants as compared to healthy plants.

### 2.9. Analysis of SOD and APX Activity

During the photosynthesis process, internal rate of O_2_ is high and consequently the by-product of this mechanism produces several O^2−^ free radicals. These free radicals lead to oxidative stress in the cell. Hence, for protection from oxidative stress, cells are equipped with antioxidant defense system. Therefore, we analyzed the antioxidant enzyme activity, particularly APX and SOD activity, in healthy and infected papaya leaves. The APX activity was found to be significantly decreased by 56.88% in PaLCuV-infected leaves (Figure 9E). Likewise, the SOD activity was also declined by 85.27% in PaLCuV-infected leaves (Figure 9F). Low APX enzyme activity and SOD activity in PaLCuV-infected leaves suggested that more oxidative damage might have occurred due to ROS formation in the chloroplast. Similar trends of results have been observed by several other researchers involved in both biotic and abiotic stress studies [51].

### 2.10. Analysis of Catalase Activity

CAT enzyme acts as a monitoring index for plant response to viruses. The catalase activity was significantly reduced (74.49%) in PaLCuV-infected leaves (Figure 9G). The reduction in CAT activity may have been due to inhibition of the enzyme substrate—H_2_O_2_ [54]. Catalase dismutates hydrogen peroxide (H_2_O_2_) into H_2_O and O_2_. The reduced catalase activity was also observed by several other researchers in different plants infected with viruses [51,55,56].

### 2.11. Soluble Sugar Analysis

Carbohydrates, which represent one of the main organic constituents of the dry matter, are derived from photosynthesis. To understand the changes of carbohydrate pattern, we conducted the soluble sugar analysis in healthy and PaLCuV-infected leaves. The results showed that PaLCuV-infected plants exhibited markedly decreased total soluble sugar as compared with healthy plants. The magnitude of the decrease in total soluble sugar contents was 70.37% as compared to the reference healthy plants (Figure 9H). A similar trend of results was also obtained by Radwan et al. [57]. They accounted that the contents of chlorophyll as well as carbohydrates (including both insoluble and soluble carbohydrates) were decreased in *Cucurbita pepo* leaves infected with zucchini yellow mosaic virus (ZYMV). The reduction in soluble sugar in PaLCuV-infected plants may have either been due to impairment of the photosynthesis apparatus (which causes decreases in photosynthetic activity) and/or due to increased respiration [58].

### 2.12. Carica Papaya Extract (CPE) Extends C. elegans Lifespan

*C. elegans* is an important model organism for studying drug screening. To determine whether different pharmacological dosages, *viz*., 50 µg/mL, 100 µg/mL, and 200 µg/mL of CPE would affect the mean life span of *C. elegans* at 20 °C, we conducted a life span experiment. The result presented in Table 1 clearly showed that different concentrations of CPE extended the mean life span of *C. elegans*. A significant increment of 29.7% in mean life span was obtained at 100 µg/mL dosage of the extract, followed by 50 µg/mL (10.4% increase) and 200 µg/mL (10.4%) compared to untreated control animals (Table 1, Figure 10). Thus, maximum increase in the lifespan was recorded at 100 µg/mL CPE dosage. It was also previously reported that the lifespan extension in *C. elegans* is dosage-dependent [59,60,61], and a particular dosage will be more impactful than in comparison to other dosages. Our study is on par with the previous reports that a particular dose was effective in increment of maximum lifespan (100 µg/mL CPE), which was further used for comparative study with crude extract from the infected leaf sample. When the effective dose (100 µg/mL) of CPE-infected leaves was tested in comparison with 100 µg/mL healthy CPE, we observed a marginal increase of lifespan extension (18.4%). However, it was found to be lower than that of the healthy plant leaf extract at the same effective dose (100 µg/mL). This might have been due to the reduced quality of the plant extract, which was severely impacted by infection of PaLCuV.

## 3. Materials and Methods

### 3.1. Collection of Plant Samples

The papaya plants were maintained under protected condition at ICAR-Central Institute for Subtropical Horticulture, Lucknow, India (26°45′–27°10′ N latitude, 80°30′–80°55′ E longitude). The area is located at 123 m above sea level and received average rainfall of 1000 mm (https://en.climate-data.org/asia/india/uttar-pradesh/Lucknow, accessed on 18 December 2021). The seedlings of papaya variety Pusa Delicious were grown under greenhouse conditions in the month of March and were kept for observation. After 3 months, one set of plants (5 plants) was infected with white fly under controlled conditions. Further, after 48 h of acquisition/inoculation period with viruliferous whiteflies, plants developed symptoms in two to four weeks’ time. After the infection stage, the whiteflies were neutralized by application of an insecticide. Upon development of typical PaLCuV symptoms, the infected leaves as well as healthy leaves from non-infected plants were collected and immediately transferred to the laboratory for further experimental studies.

### 3.2. Identification of PaLCuV Infection in Papaya Leaves

For the identification of PaLCuV infection, we performed PCR analysis using a set of degenerate primers—PAL1V-1978 and PAR1C-715—to detect the presence of PaLCuV (Table 2). The PCR conditions were as follows: 94 °C-5 min; 94 °C-30 s; 56 °C-90 s; 72 °C-120 s (35 cycles); 72 °C-5 min. The amplification of a 1.6 kb band on 0.8% agarose gel confirmed the presence of a geminivirus causing leaf curl disease in papaya.

Total DNA was extracted from 1 g of healthy and PaLCuV-infected papaya leaves using the cetyl trimethyl ammonium bromide (CTAB) method [62].

**Table 2 plants-11-00579-t002:** Primer sequence used for molecular characterization of PaLCuV-infected leaves.

Gene Name	Fwd/Rev	Nucleotide Sequence	References
*PAL1V1979*	Fwd	GCATCTGCAGGCCCACATYGTCTTYCCNGT	Bela-ong and Bajet, 2007 [63]
*PAR1C715*	Rev.	GATTTCTGCAGTTDATRTTYTCRTCCATCCA	Bela-ong and Bajet, 2007 [63]

### 3.3. Study on Alteration in Leaf Anatomy

#### Leaf Stomatal Density and Guard Cell Size

The leaf stomatal density was estimated as per the previous protocols reported in our manuscript [64]. The density was measured by using the impression approach method, which is expressed as the number of stomata per unit leaf area [65]. The lower surface was cleaned and smeared at the mid area of the leaf. A very fine layer denuded from the leaf surface was then mounted on the glass slide and utilized for the estimation of the stomatal density. Ten samples from each healthy and infected plant were observed and recorded using a light microscope (MPS 60, Leica, Wetzlar, Germany). The density of stomata in the leaf was calculated using following formula [64]:$$\mathrm{Stomatal}\;\mathrm{density}\;=\frac{\mathrm{Numbers}\;\mathrm{of}\;\mathrm{stomata}}{\mathrm{Area}\;\left(0.072463{\;\mathrm{mm}}^{2}\right)}$$

The stomatal size was defined as the length (in µm) between the junctions of the guard cells at each end of the stomata and was taken to indicate the maximum potential that revealed the maximum opening potential of stomatal pore [64,66].

### 3.4. Study on Alteration in Physiological Properties

#### 3.4.1. Photosynthesis, Transpiration, Stomatal Conductance, and Intercellular CO_2_ Concentration Measurement

The papaya leaves from both healthy and infected plants were taken for observation on physiological parameters, *viz*, net photosynthesis rate (A), transpiration (E), stomatal conductance (gs), and intercellular CO_2_ concentration. The portable photosynthetic system (Li-Cor, Lincoln, NE, USA) was used for observation of the aforementioned parameters. The CO_2_ levels inside the leaf cuvette were maintained at 400 ppm, photosynthetic photon flux density (PPFD) was 600 μmol m^−2^ s^−1^, leaf temperature was 30 °C, and leaf–air vapor pressure deficit was kept at <3.0 kPa.

#### 3.4.2. Estimation of Plant Pigments (Chlorophyll a, Chlorophyll b, Carotenoid, and Anthocyanin)

The plant pigments were estimated by following the protocol described by Wellburn [67]. The known quantity of plant leaf samples was taken and suspended in 10 mL (80%) acetone solvents. The samples were shaken well and kept overnight (4 °C) in the dark. Afterwards, samples were centrifuged (5000× *g*) at 4 °C for 5 min, and the supernatant was collected, followed by absorbance at 663 nm, 645 nm, and 450 nm.

The plant pigments, including both types of chlorophyll and the carotenoids, were calculated (in mg g^−1^ of FW of leaf tissue) by the following formulae:Chl a = {(0.0127 × A_663_)} − (0.00269 × A_645_)} × 1000
Chl b = {(0.0229 × A_645_)} − (0.00468 × A_663_)} × 1000
Crtd = {(4.07 × A_450_)} − (0.0435 × Chl a) + (0.367 × Chl b)}
where Chl a = chlorophyll a; Chl b = chlorophyll b; Crtd = carotenoids; A_663_ = absorbance at 663 nm; A_645_ = absorbance at 645 nm; A_450_ = absorbance at 450 nm.

For the quantification of anthocyanin, we extracted the samples in 1% acidified methanol solvents instead of acetone. The absorbance of collected supernatant was observed at 530 and 650 nm. Further, the corrected value of anthocyanin absorbance was calculated by using following formula:AA = A_530_ − (0.288 × A_650_)
where AA is corrected anthocyanin absorbance; A_530_ = absorbance at 530 nm; A_650_ = absorbance at 650 nm.

The total anthocyanin content in samples was calculated using corrected absorbance (AA) and a molar absorbance coefficient for anthocyanin at 530 nm of 30,000 L mol^−1^ cm^−1^ [68].

#### 3.4.3. Chlorophyll Fluorescence and P700 Measurements of Healthy and Virus-Infected Leaves

The MODULAR Version of DUAL-PAM (DUAL-PAM-100, a P700 and chlorophyll fluorescence measuring system) equipped with DUAL-DB detector and accessory P700 dual-wavelength emitter (WALZ, Effeltrich, Germany) was used for simultaneous detection of the P700 oxidized state as well as for chlorophyll fluorescence of photosystem II. The DUAL-PAM software was used to analyze the chlorophyll fluorescence and other parameters such as effective quantum yield (photochemical energy use, denoted by Y) of photosystem I (PSI), i.e., YI, and photosystem II (PSII), i.e., YII [68,69]; non-photochemical quenching of chlorophyll fluorescence (NPQ); proportion of energy spread out in form of heat via the regulated non-photochemical quenching mechanism, i.e., regulated non-photochemical loss (Y(NPQ)); proportion of energy passively spread out in form of heat and fluorescence, i.e., non-regulated non-photochemical loss (Y(NO)) [32]; non-photochemical loss due to oxidized primary donor (Y(ND)); and non-photochemical loss due to reduced acceptor (Y(NA)).

The electron transport rate (ETR) of PSI, i.e., ETR(I), and of PSII, i.e., ETR(II), were calculated as follows:ETR(I) = 0.5 × Y(I) × 0.84 × PPFD
ETR(II) = 0.5 × Y(II) × 0.84 × PPFD
where the value of 0.84 was assumed to absorb 84% of incident photons.

The value of 0.5 was denoted, meaning that only 50% photons were absorbed by the photosystem (i.e., proportion of light reaching to photosystem).

PPFD represents the photosynthetic photon flux density, i.e., sum of active photons (4000–7000 A°) striking per unit time on a unit surface.

The maximal PSII quantum yield was calculated as follows:$$\mathrm{The}\;\mathrm{maximal}\;\mathrm{PSII}\;\mathrm{quantum}\;\mathrm{yield}\;=\frac{\mathrm{Fm}-\mathrm{Fo}\;}{\mathrm{Fm}}=\frac{\mathrm{Fv}\;}{\mathrm{Fm}}$$
where

Fm is the maximal fluorescence yield of dark-adapted sample with all PSII centers closed.

Fo is the minimal fluorescence yield of dark-adapted sample with all PSII centers open.

Fv/Fm is determined after exposure to the dark for about 20 min [69].

The ETR(I) > ETR(II) when cyclic electron flow (CEF) is activated. Therefore, difference in their value, i.e., (ETR(I) − ETR(II)) was the calculated value of CEF [68,69].

The redox state P700 (chlorophyll reaction center of PSI) was monitored using DUAL PAM-100 with a dual wavelength (830/870 nm) [69]. A saturation pulse (10,000 μmol m^−2^ s^−1^) was applied for the measurement of chlorophyll fluorescence as well as assessment of P700 parameters. The P700+ signals may vary between maximal (completely oxidized state of P700) and minimal (completely reduced state of P700) levels. The completely oxidized state, i.e., maximal level (analogy to Fm and denoted as Pm) was monitored with application of a saturation pulse after pre-illumination with far-red light. The Pm’ (analogy to Fm’, i.e., maximum fluorescence during actinic light illumination) and P700 parameters were determined with background actinic light (830 μmol m^−2^ s^−1^) exposure for 10 min.

The photochemical quantum yield of PSI (Y(I)) was defined by the proportion of overall P700 that in a given state is reduced and not limited by the acceptor side. Y(I) was calculated as
Y(I) = 1 − Y(ND) − Y(NA).
where Y(ND) represents the proportion of overall P700 that is oxidized in a given state, and Y(NA) represents the proportion of overall P700 that cannot be oxidized by a saturation pulse in a given state due to a lack of oxidized acceptors.

#### 3.4.4. Measurements of Free Proline Content

The total free proline content was estimated by following the method of Bates et al. [70]. A total of 100 mg of healthy and PaLCuV-infected plant leaf samples were mixed in 1.5 mL of 3% (*w*/*v*) sulphosalycylic acid. The samples were properly mixed and homogenized, followed by filtering using Whatman filter paper. Afterwards, 100 μL of filtrate was taken and mixed with glacial acetic acid (2 mL) and acid ninhydrin (2 mL). The resulting cocktails were heated in a water bath for 1 h (at 100 °C), followed by quick cooling in an ice bath in order to terminate the reaction. The cooled blended samples were extracted using toluene and left at room temperature (25 °C) until the temperature was normalized to peripheral environment, followed by absorbance being taken at 520 nm. The concentrations of proline in samples were estimated using a standard curve.

#### 3.4.5. Lipid Peroxidation Assay

For malondialdehyde (MDA) and thiobarbituric acid (TBA) estimation, 100 mg of healthy and PaLCuV-infected plant leaf samples were homogenized in chilled extraction buffer (Na_2_HPO_4_·12H_2_O (1.6%) + NaH_2_PO_4_·2H_2_O (0.7%)) using a pestle and mortar. Afterwards, the blended samples were centrifuged (15,000× *g* for 10 min at 4 °C), and a supernatant (0.5 mL) was mixed with 1.5 mL TBA (0.5%) containing 20% trichloro acetic acid (TCA). The reaction cocktail was incubated in a water bath for 25 min (at 95 °C), followed by quick cooling in an ice bath in order to seize the reaction. The optical density (OD) of samples was recorded at 532 and 600 nm. After subtracting the non-specific absorbance value (value obtained at OD600 − value obtained at OD532), we calculated the MDA concentration by its molar extinction coefficient (€^M^ = 155 mM^−1^ cm^−1^) [71,72].

#### 3.4.6. Estimation of Relative Electrolyte Conductivity (REC) and Osmotic Potential

The electrolyte conductivity of plant leaves was estimated using a conductivity meter [73]. The 10 mm^2^ long leaf discs were sliced from healthy and PaLCuV-infected leaves. The samples were kept in 10 mL of distilled water (in test tube) for 4 h, and afterwards, conductivity of the solution (L1) was recorded. Thereafter, the samples were autoclaved. The conductivity of autoclaved samples was also recorded (L2). The relative water conductivity (REC) was calculated by the following formula:$$\mathrm{REC}\;\left(\%\right)=\frac{L1}{L2}\times 100$$

#### 3.4.7. Ascorbate Peroxidase (APX) Assay

The APX activity was determined using modified protocol of Nakano and Asada [74]. For the assaying of APX activity, the homogenized reaction cocktail (3 mL), containing crude extracts of leaf along with potassium phosphate buffer (50 mM, pH 7.0), ascorbic acid (0.5 mM), EDTA (0.1 mM), hydrogen peroxide (0.1 mM), APX, and water, was prepared. The enzyme activity was determined by recording the progressive decrease in absorbance (due to oxidation of ascorbate in the reaction mixture) using a spectrophotometer at 290 nm. The reaction was run for 5 min (at 25 °C). The enzyme necessary for decomposition of 1 μmol ascorbate min^−1^ was referred to as one unit.

#### 3.4.8. Superoxide Dismutase (SOD) Activity

The homogenized reaction cocktail (prepared in pre-chilled mortar), containing leaf samples (250 mg) in potassium phosphate buffer (100 mM, pH 7.8), EDTA (0.1 mM), triton-X 100 (0.5% *v*/*v*), and polyvinyl pyrrolidone (1% *w*/*v*), was prepared. The Bradford dye binding method was used for protein quantification [75].

For assaying SOD activity, the 20 µL protein extracts were added into a reaction blend of phosphate buffer (100 mM), Nitroblue tetrazolium (NBT (57 mM)), L-methionine (10 mM), and triton-X 100 (0025%, *v*/*v*). Thereafter, riboflavin (4.4%, *w*/*v*) was added. The final prepared reaction cocktail developed color on exposure of photon (µmol m^–2^ s^–1^) for 7 min. The non-photon exposed reaction cocktail (not developed color) served as a control. The absorbance of both samples and control were recorded at 560 nm. The final absorbance of samples was calculated by subtracting its absorbance from the control. The amount of enzyme necessary for 50% inhibition of rate of NBT reduction was referred to as one unit [36].

#### 3.4.9. Estimation of Catalase (CAT) Activity

The catalase activity in both healthy and PaLCuV-infected leaves was estimated by following the protocol described by Hameed et al. [76]. One gram of leaf sample was homogenized with a reaction mixture containing sodium phosphate buffer (100 mM, pH 7.0), triton-X 100 (1% *v*/*v*), and 2-mercaptoethanol (7 mM). Afterwards, the homogenized samples were centrifuged (12,000× *g* for 20 min at 4 °C), and supernatants were collected. The CAT activity was estimated by preparing assay solution (3 mL) containing supernatant (0.1 mL), hydrogen peroxide (5.9 mM), and phosphate buffer (0.1 mL, 50 mM), followed by observation of absorbance at 240 nm.

#### 3.4.10. Soluble Sugar

The estimation of soluble sugars was performed by following the method described by Scott and Melvin [77]. One gram of both healthy and infected (dried) tissues was homogenized with ethyl alcohol. The homogenates were boiled (15 min) in a water bath, which was followed by filtration (after samples were cooled) using Whatman filter paper. The filtrate thus obtained was dried in an oven (65–80 °C). Thereafter, the known volume of water was added to the samples for determination of soluble sugar. The quantity of estimated soluble sugar was expressed in μg 100 mg^−1^ Distill water (DW).

### 3.5. Study on Alteration in Bioactive Properties

#### 3.5.1. Extract Preparation of Papaya Leaves

One hundred grams of fresh *C. papaya* leaves was grounded into 200 mL of distilled water in an electronic grinder. The sample was centrifuged (2000× *g* for 5 min) at room temperature. The supernatant thus obtained was filtered through fine muslin cloth and stored at 4 °C until further use [78].

#### 3.5.2. *Caenorhabditis elegans* Growth Condition and Extract Treatment

The wild-type *C. elegans* (N_2_ Bristol) was maintained at 20 °C under standard laboratory conditions on nematode growth media (NGM) with *E. coli* OP50 as a food source [79]. *C. elegans* strain as well as the OP50 strain were obtained from the Caenorhabditis Genetics Centre (CGC), University of Minnesota, USA [59,60]. Different pharmacological dosages (50 µg/mL, 100 µg/mL, and 200 µg/mL) of *C. papaya* extract (CPE) were prepared and added directly to the *E. coli* OP50 food source to feed the worms on it.

#### 3.5.3. Lifespan Assay

Age-synchronized worms were used for lifespan assay. Isolated eggs were allowed to hatch on NGM plates with or without different concentrations of CPE. At L4 stage, 25 worms were transferred to 30 mm fresh NGM plates containing corresponding test concentrations of CPE. Worms were transferred to a fresh plate every day until the post-fertile stage, and after that, every 2–3 days to assure the presence of CPE throughout the experiment until they were scored as dead. Survival of worms was scored every day with a platinum wire until the last worm was dead [59,60].

## 4. Conclusions

Papaya an important crop, not only for the horticulture industry, but also for the pharmaceutical industry. Papaya leaf extract is used by the pharmaceutical industry for various ailments. Papaya leaf curl virus disease (PaLCuV) is an important phytopathogen that is responsible for low-quality raw material, one that is significantly impacting the quality production of leaves. It is therefore imperative to study the impact of disease on physiological and biochemical parameters of papaya leaves and their pharmacological properties so that a suitable mitigation strategy can be developed. Our study clearly revealed that healthy papaya leaf extract enhanced the life span of the *C. elegans* model, whereas the infected leaf extract showed decreased efficacy. Since papaya leaf is an important choice for the wellness industry, it is imperative to have healthy raw material for effective and active supplement. Indeed, it is almost necessary to have due attention paid to sustainable management strategies for PaLCuV disease in order to maintain the quality of papaya leaf and losses of active properties.

## Figures and Tables

**Figure 1 plants-11-00579-f001:**
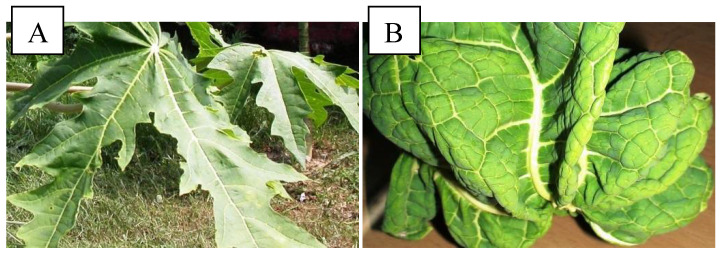
Morphology of (**A**) healthy and (**B**) PaLCuV-infected leaf.

**Figure 2 plants-11-00579-f002:**
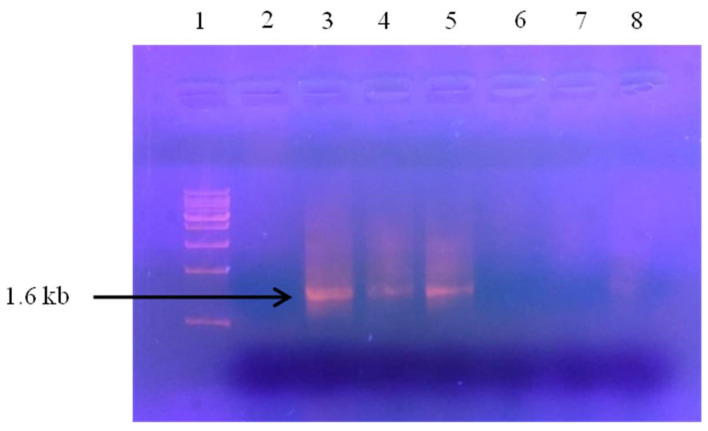
PCR detection of papaya leaf curl disease in papaya leaf samples: Lane 2 = healthy control leaf showing no symptoms. Lanes 3, 4, and 5 are symptomatic papaya leaf samples. The amplicon with band size of ≈1.6 kb in Lanes 3, 4 and 5 shows presence of PaLCuV. Lane 1 is a 250 base pair step up ladder (Genei, India). Lanes 6, 7, and 8 display leaf samples from asymptomatic field plants showing no amplification.

**Figure 3 plants-11-00579-f003:**
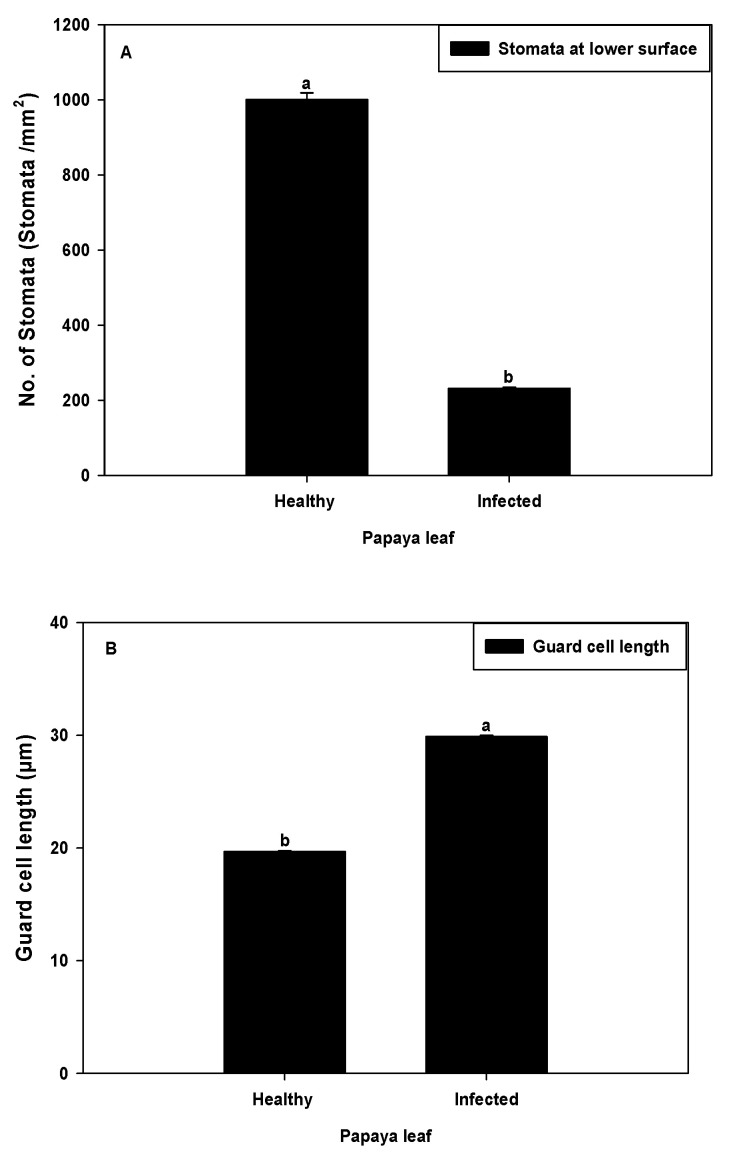
(**A**) Comparative analysis of stomatal density of healthy and infected papaya leaf. (**B**) Comparison of guard cell length of healthy and infected papaya leaf. Error bars shown as the standard error of mean (SE) was computed by Sigma Plot 14, different letters above the error bars show significant differences at *p* ≤ 0.05.

**Figure 4 plants-11-00579-f004:**
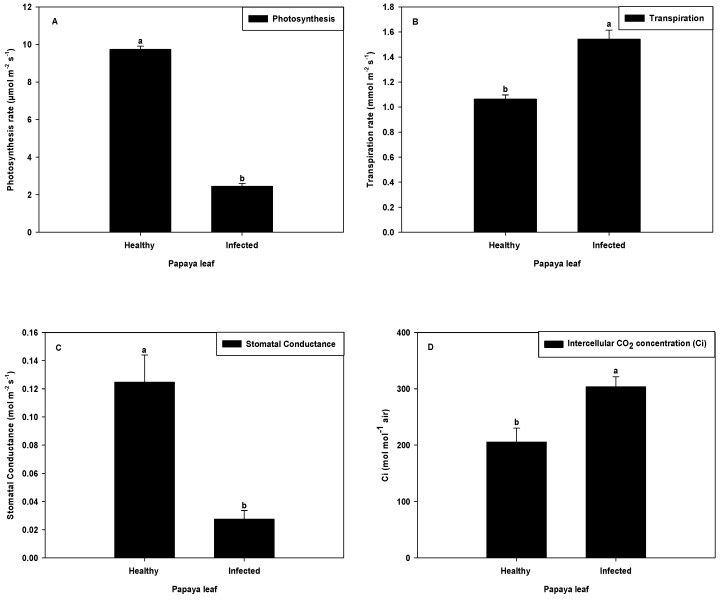
Effect of PaLCuV virus infection on (**A**) photosynthesis rate, (**B**) stomatal conductance, (**C**) transpiration rate, and (**D**) intercellular CO_2_ concentration. Error bars shown as the standard error of mean (SE) was computed by Sigma Plot 14, different letters above the error bars show significant differences at *p* ≤ 0.05.

**Figure 5 plants-11-00579-f005:**
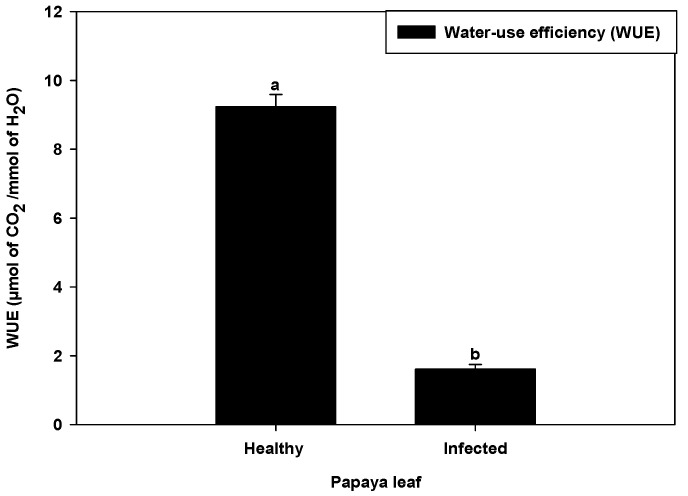
Water use efficiency in papaya leaves. Error bars shown as the standard error of mean (SE) was computed by Sigma Plot 14, different letters above the error bars show significant differences at *p* ≤ 0.05.

**Figure 6 plants-11-00579-f006:**
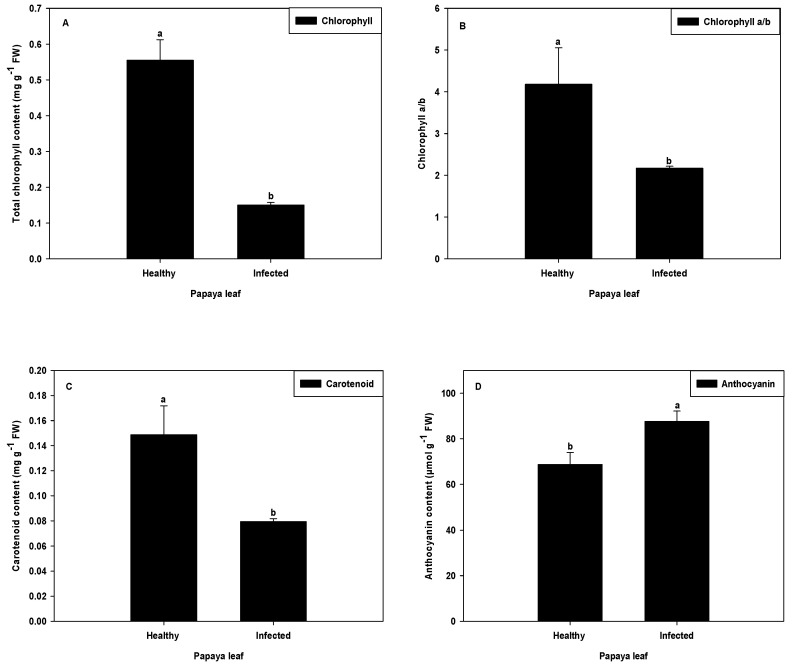
Effect of PaLCuV virus infection on (**A**) total chlorophyll content, (**B**) chlorophyll ratio, (**C**) carotenoid content, and (**D**) anthocyanin content. Error bars shown as the standard error of mean (SE) was computed by Sigma Plot 14, different letters above the error bars show significant differences at *p* ≤ 0.05.

**Figure 7 plants-11-00579-f007:**
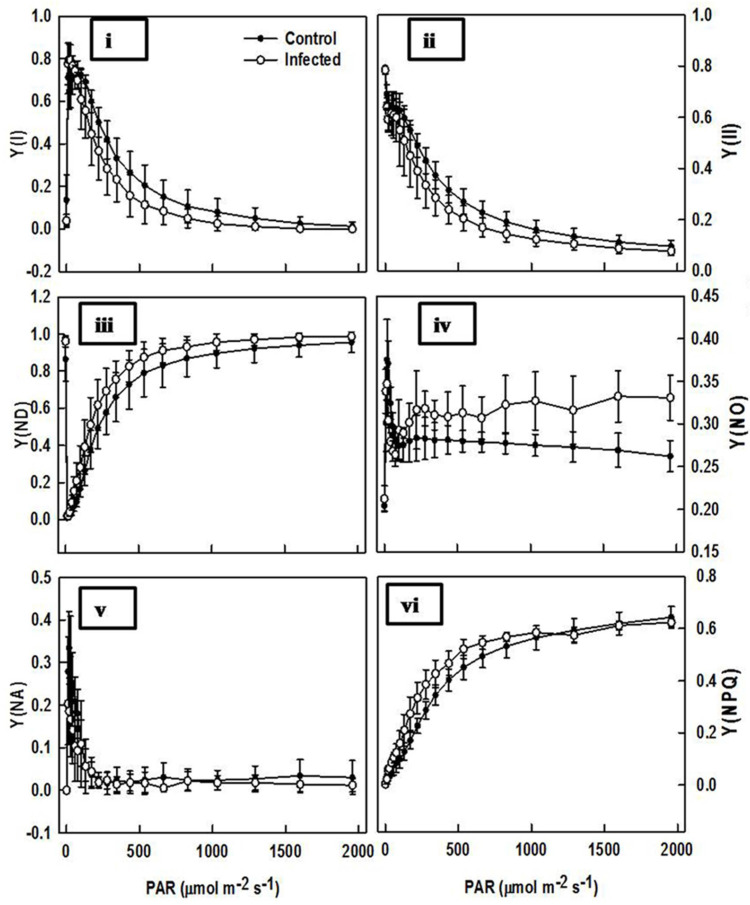
Light response of PSI and PSII in control (healthy) (-●-) and leaf curl-infected (-o-) leaves. (**i**) Y(I): Photochemical quantum yield of PSI. (**ii**) Y(II): Effective quantum yield of PSII. (**iii**) Y(ND): Quantum yield of non-photochemical energy dissipation due to donor side limitation. (**iv**) Y(NO): Quantum yield of non-regulated energy dissipation in PSII. (**v**) Y(NA): Quantum yield of non-photochemical energy dissipation due to acceptor side limitation. (**vi**) Y (NPQ): Quantum yield of regulated energy dissipation in PSII.

**Figure 8 plants-11-00579-f008:**
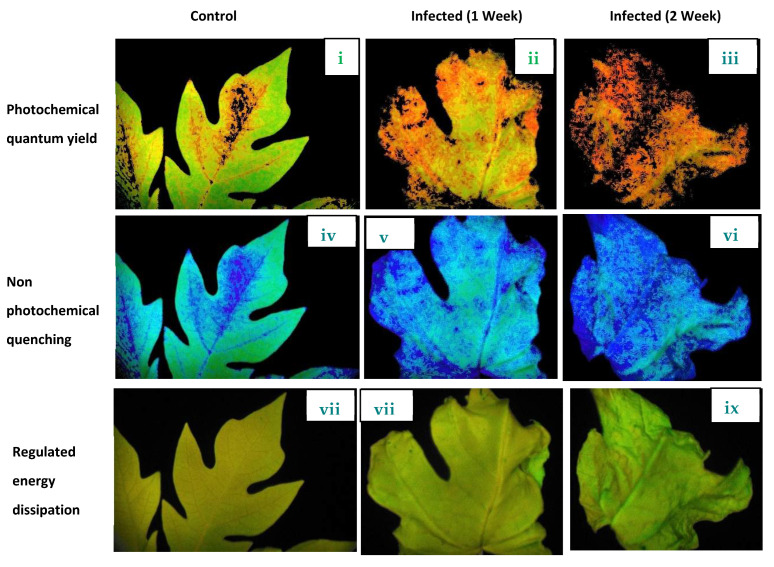
Imaging PAM chlorophyll fluorometer measurements of healthy plant leaf and diseased leaf. The false color scale depicted to the right of images shows the amplitude of the particular parameter. (**i**–**iii**) Comparison of quantum yield (YII). (**iv**–**vi**) Non-photochemical quenching (NPQ). (**vii**–**ix**) Regulated energy dissipation of healthy, mild-stage, and advanced-stage PaLCuV-infected leaves of papaya.

**Figure 9 plants-11-00579-f009:**
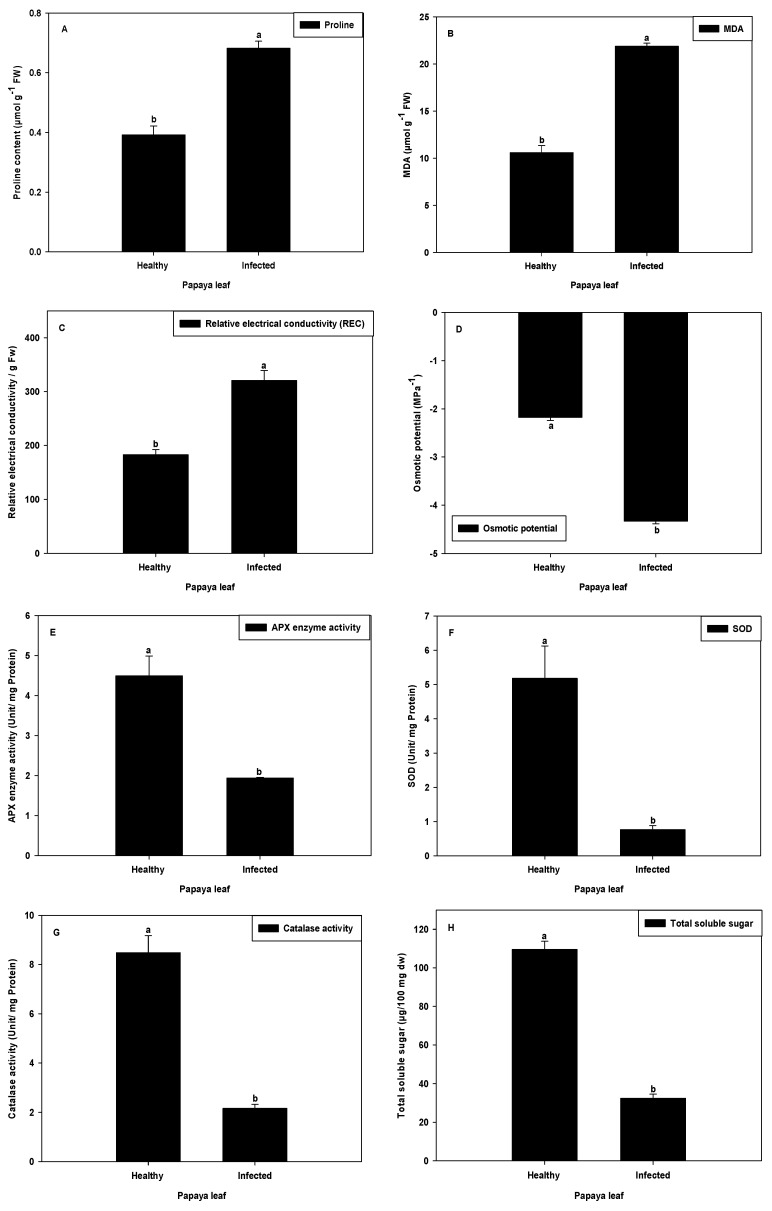
Effect of PaLCuV virus infection on (**A**) proline content, (**B**) MDA, (**C**) relative electrical conductivity, (**D**) osmotic potential, (**E**) APX enzyme activity, (**F**) SOD, (**G**) catalase activity, and (**H**) total soluble sugar. Error bars shown as the standard error of mean (SE) was computed by Sigma Plot 14, different letters above the error bars show significant differences at *p* ≤ 0.05.

**Figure 10 plants-11-00579-f010:**
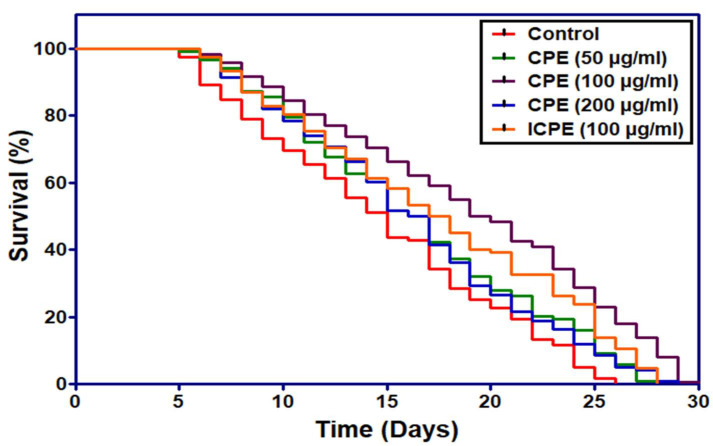
Effect of papaya leaf extract supplementation on lifespan of wild-type *C. elegans*.

**Table 1 plants-11-00579-t001:** Effect of *C. papaya* leaf extract on the lifespan of *C. elegans* at 20 °C.

N_2_ Wild Type	Mean Lifespan ± SE	Percent Change	Max. Lifespan ± SE
Control	14.76 ± 0.3		25 ± 0.4
CPE 50 µg/mL	16.28 ± 0.1	10.4	27 ± 0.4
CPE 100 µg/mL	19.11 ± 0.1	29.7	29.2 ± 0.2
CPE 200 µg/mL	16 ± 0.3	10.4	27.2 ± 0.7
ICPE 100 µg/mL	17.46 ± 0.2	18.4	27.7 ± 0.2

## Data Availability

Not applicable.

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
