# Peer review of "Papaya Leaf Curl Virus (PaLCuV) Infection on Papaya (*Carica papaya* L.) Plants Alters Anatomical and Physiological Properties and Reduces Bioactive Components"

_plants, 2022, doi:10.3390/plants11050579_

Round 1

Reviewer 1 Report

The manuscript, even if presents several interesting biochemical experiments between healthy and virus-infected papaya extracts, suffers for a certain inaccuracy in writing and the way results are presented.

I feel a different English quality in writing among several pieces: so an accurate revision is needed: 69 (are occurred); 507 (responsible low quality); 347 (lowwer); 350 (recoreded ); 346 (denisty); 175 (are accordance); 204 (did not had ) , 314-316 (comprasion to other dosases , etc...);

some other points that require  attention :

63: Bemisia tabaci

325-329: it is very important to describe the age and condition of plants (potted in screenhouse or thermo-conditioned greenhouse? seedlings ? clonally propagated and grafted on a seedling rootstock ? etc); protected condition does not mean that rainfall is the only source of watering...

330, 82-86: how long is the symptom latency period (from virus transmission by vectors up to early evidence)? months , days  ?

could the author describe concentration of virus by qPCR at time of biochemical experiments or during development of symptoms ? so not using a diagnostic amplicon as long as 1600bp !

have the experiments been done in a short temporary windows ?

388, 421: CEF or MDA, please before using acronyms explain the full spelling of abbreviations (here and in other sentences)

Fig 2. , 96: you detect a virus genome and not a 'disease'

101: specify , if possible, age of the leaves at what sampling time-point of infection

149-150: there is no picture at microscope which substantiates this statement

197-204: these sentences need a deep revision: try to rephrase and when needed support with references the statements

260: proline

par. 2.12: it is absolutely unclear to me the role of infected versus healthy leaf sap extracts in C. elegans trial ( at the same dose ). The evidences in the paragraph should be better formulated: " In comparison to 100 μg/ml CPE, same concentration of CPE is less effective (18.4% increase)" ; is the second CPE meaning the infected one ??

Author Response

Subject: Revised manuscript "Papaya Leaf Curl Virus (PaLCuV) infection on papaya (Carica papaya L.) plants alters anatomical and physiological properties and reduces bioactive components. Plants 1538980 R1.

Dear Reviewer,

We are pleased to resubmit the revised version of our paper 1538980 R1. First of all, we would like to thanks to the reviewer for the time he spent in reviewing our manuscript and for their wise comments to improve overall quality of the manuscript.

We have made all the corrections/changes suggested by the reviewer. All the authors have invested sufficient time in revising the manuscript and fix all the errors as suggested.  All authors are agree for submission of revised version of the manuscript.

Reviewer#:1

The manuscript, even if presents several interesting biochemical experiments between healthy and virus-infected papaya extracts, suffers for a certain inaccuracy in writing and the way results are presented.I feel a different English quality in writing among several pieces: so an accurate revision is needed: 69 (are occurred); 507 (responsible low quality); 347 (lowwer); 350 (recoreded ); 346 (denisty); 175 (are accordance); 204 (did not had ) , 314-316 (comprasion to other dosases , etc...);

Ans-The manuscript has been critically revised by all the authors and errors have been fixed. All the suggested corrections were included and highlighted in the MS.

some other points that require  attention : 63: Bemisia tabaci

Ans- Corrected

325-329: it is very important to describe the age and condition of plants (potted in screenhouse or thermo-conditioned greenhouse? seedlings ? clonally propagated and grafted on a seedling rootstock ? etc); protected condition does not mean that rainfall is the only source of watering...

Ans- Papaya is propagated through sexual means. Therefore, seedlings of papaya variety Pusa Delicious was grown under green house in the month of March. The plants were kept for 3 months for observation. Infected whiteflies were released on the green house on 3 months old plants.  The explanation is included in the methods section (3.1).

330, 82-86: how long is the symptom latency period (from virus transmission by vectors up to early evidence)? months, days  ?

Ans- After 48 hrs of acquisition/ inoculation period with viruliferous whiteflies, plants developed symptoms in two to four weeks time. The explanation has been included in the methods section (3.1).

could the author describe concentration of virus by qPCR at time of biochemical experiments or during development of symptoms ? so not using a diagnostic amplicon as long as 1600bp !

Ans-Thank you for the quarry, we have conducted experiments on full development of symptoms which was visually observed. We just detected the presence of virus for confirmation but didn’t included any experiments for virus load during different time periods of the experiment.

have the experiments been done in a short temporary windows ?

Ans- No, experiments were conducted during the entire plant phase. The experiment was also repeated once.

388, 421: CEF or MDA, please before using acronyms explain the full spelling of abbreviations (here and in other sentences)

Ans- Done and included in the MS

Fig 2. , 96: you detect a virus genome and not a 'disease'

Ans- We are sorry for the confusing sentence, the disease has been already reported. We infected with virus and thus reconfirmed the presence of PaLCuV. So PCR amplification of PaLCuV gene was done for confirmation of their presence only. 

101: specify, if possible, age of the leaves at what sampling time-point of infection

Ans- The young as well old leaves having typical symptoms of leaf curl were selected for the study

149-150: there is no picture at microscope which substantiates this statement.

Ans- The SEM photograph has already been published earlier as a part of this research which was included in the MS. (Mishra M.K, Mishra M, Kumari S, Shirke P, Srivastava A, Saxena S (2019) Studies on anatomical behaviour of PaLCuV infested papaya (Carica papaya L.) J Appl Hort DOI: 10.37855/jah.2018.v20i03.38 )

197-204: these sentences need a deep revision: try to rephrase and when needed support with references the statements

Ans- Thank you for the suggestions. The paragraph in the section 2.5 (197-204) has been worked in and rephrased accordingly.

260: proline

Ans-Done

par. 2.12: it is absolutely unclear to me the role of infected versus healthy leaf sap extracts in C. elegans trial (at the same dose ). The evidences in the paragraph should be better formulated: " In comparison to 100 μg/ml CPE, same concentration of CPE is less effective (18.4% increase)" ; is the second CPE meaning the infected one ??

Ans- The reviewer is right the experiment was first conducted with healthy CPE, leaf extract, and we found 100 μg/ml CPE was most effective in life span extension. Further the same dose was taken from infected crude extract and assessed for bio-potentials.  The paragraph has been rephrased, and included in the mentioned section.

All the corrections/changes in revised manuscript have been highlighted in red colour. Based on all of the above, we are confident that the revised version of our manuscript is suitable for publication in your esteemed journal.

Looking forward to your response

With best and warm regards

Reviewer 2 Report

This manuscript deals with the effect of viral infection on foliar morphology and physiology of Carica papaya. The topic is within the scope of the journal, is interesting and original. This very complete study is of great interest because of its contribution to the knowledge of modifying the health and properties of papaya leaves when infected.

The methodology used is correct and properly justified, and the results are sound. However, the description of the methodology is incomplete since the experimental design, the sample size and the type of sampling (random, systematic, ...) are not described. Likewise, neither the statistical model applied to the data is described, nor the level of significance considered (p-value) to admit significant differences between treatments. The manuscript is well structured and written. Although I do not feel qualified to evaluate the quality of English it must be revised, mainly the lack of some punctuation marks (periods, commas). The bibliography is very up-to-date. So, there are some aspects of the description of Material and Methods that should be reviewed. All this and the particular comments are highlighted in color in the attached file.

Therefore, I recommend this manuscript for publication but after a major revision taking into account the reviewer comments.

Author Response

Subject: Revised manuscript "Papaya Leaf Curl Virus (PaLCuV) infection on papaya (Carica papaya L.) plants alters anatomical and physiological properties and reduces bioactive components. Plants 1538980 R1.

Dear Reviewer,

We are pleased to resubmit the revised version of our paper 1538980 R1. First of all, we would like to thanks to the reviewer for the time they have spent in reviewing our manuscript and for their wise comments to improve overall quality of the manuscript.

We have made all the corrections/changes suggested by the reviewers. All the authors have invested sufficient time in revising the manuscript and fix all the errors as suggested.  All authors are agree for submission of revised version of the manuscript.

Reviewer 2:

Qns. L22, Qns. L24; APX, SOD and CAT; As this is the first time these acronyms have appeared in the text, they must be written in full words.

Ans. DONE. Add as per suggestion.

Qns. L24; MDA; Please, type it in full.

Ans. DONE. Add as per suggestion.

Qns. L28-L29; Fv/Fm, Y (I), Y(II), Y(NPQ); As this is the first time these acronyms have appeared in the text, they must be written in full words.

Ans. DONE. Add as per suggestion.

Qns. L32; @ ??

Ans. @ is replaced by appropriate words.

L42; A period "." is missing.

DONE. Add as per suggestion.

L56; Replace the period with a comma.

DONE. Add as per suggestion.

L123; missing period.

DONE. Add as per suggestion.

L126; comma missing.

DONE. Add as per suggestion.

L141; The transpiration rate (E); What do the authors think may be the reason why stomatal conductance decreases in infected plants while transpiration increases? A comment would be welcome.

Ans- Thank you for the quarry, we have been working with the abiotic stress, where phosphorus starvation was done in chickpea maize and wheat. We found that interesting results that stomata activity is severely impacted while that plants were oozing out more water/ transpiration. We published research in New Phytologist (Direct foliar uptake of phosphorus from desert dust. Avner Gross, Sudeep Tiwari, Ilana Shtein, Ran Erel (2021). New phytologist 230: 2213–2225 https://doi.org/10.1111/nph.17344). 

Although it is not linked to virus infection but a biotic stress may have same mechanism as the abiotic stress impacting stomata activity while more transpiration.

L166-L168; The diminution in WUE may be due to reduction of stomatal numbers in infected plants, which regulate the photosynthesis and transpiration rate, consequently modulated the WUE of plants; This statement is not entirely correct, since a decrease in the number of stomata would imply less transpiration and, therefore, an increase in WUE. However, infection with the virus modified the density and size of the stomata and probably also the movement of the occlusive cells and the rate of C assimilation (decrease in photosynthesis, Fv/Fm and Y(I), and increase in Ci). All of this led to a decrease in WUE.

Ans- The question raised by the reviewer is right, however an in depth study is required to prove this so. As per previous reports (Lawson T, Blatt MR (2014) Stomatal size, speed, and responsiveness impact on photosynthesis and water use efficiency. Plant physiol 164: 1556-1570 and Murray RR, Emblow MSM, Hetherington AM, Foster GD (2016) Plant virus infections control stomatal development. Sci Rep 6: 34507. doi: 10.1038/srep34507) it was established that WUE has been influenced by the photosynthesis and transpiration rate, so we tried to predict and correlate our results with the same. However in future research we will try to understand this mechanism and will probably try answer the question raised by the reviewer. 

L170; Figure 5; The Y-axis units are not correct. They should be micromolCO2 / mmolH2O.

Ans- The suggested correction is included in the Fig 5

L170; leave; Replace by "leaves".

Ans- DONE.

L198; The Fv/Fm; values do not appear in figure 7, nor in the text.

Ans- Fv/Fm; is represented by (Y (I) and Y(II) which is included in the mentioned  section 2.6.

L260; prolein; Replace "prolein" by "proline" both, in the figure caption and in the Y-axis.

DONE.

L328; One set; How many plants?

Ans - 5 plants per set

L340; using using;  Please, remove one "using"

Ans- Done

L350; 10 samples; Are they 10 samples from 10 different leaves? Please, clarify.

Ans- Yes the reviewer is right its 10 different leaves.

L353; (0.072463 mm2); What is this number? Is it the Area value? Please, clarify.

Ans- We are sorry for the confusing lines. The reviewer is right its   it’s the area, the modification has been included in the section 3.3.1.

L361; Cotton, Cotton or papaya?

DONE. Replaced the word with papaya.

L370-372; The coefficients are not multiplied by 663nm, but by the Absorbance measured at 665nm. Please, clarify this.

Ans- The reviewer is right the coefficients are not multiplied by wavelengths (663nm and 665nm) but by the Absorbance measured values. We have made corrections in the suggested lines section 3.4.2.

All the corrections/changes in revised manuscript have been highlighted in red colour. Based on all of the above, we are confident that the revised version of our manuscript will be accepted for publication in your esteemed journal.

Looking forward to your response

With best and warm regards

Round 2

Reviewer 1 Report

The modifications produced by the authors gave a serious improvement to the manuscript. There is still a quite burden load  in the M&M paragraphs that can be reduced.

please considered further points:

88-89, "The alteration in physiology and bioactive properties of papaya leaves due to PaLCuV infection however the area remains still barely researched". A verb seems to be missing ?:

306, didn’t showed : did not show

322, remove ‘in’

334, was observed

477, seedlings were grown…

481, add that whiteflies were killed by an insecticide spray after transmission period

500, please check the correct size in kilo base pairs of the amplicon

Synthesize and avoid comments/ diffuse explanation for paragraphs 3.4.3 to 3.4.9, just telling  what kind of modifications were introduced to the methods in cited published reports.

Author Response

We are pleased to resubmit the revised version of our paper 1538980 R2. First of all, we would like to thanks to the reviewer for the time he spent in reviewing our manuscript and for their wise comments to improve overall quality of the manuscript.
We have made all the corrections/changes suggested by the reviewer. All the authors have invested sufficient time in revising the manuscript and fix all the errors as suggested. All authors are agree for submission of revised version of the manuscript.

All the corrections/changes in revised manuscript have been done as track changes. Based on all of the above, we are confident that the revised version of our manuscript is suitable for publication in the esteemed journal.

Looking forward to your response
With best and warm regards

Reviewer 2 Report

The authors have done a good job, they have taken into account the comments made by reviewers and this second version has improved considerably, in such a way that it can be accepted for publication.

Author Response

We are pleased to resubmit the revised version of our paper 1538980 R2. First of all, we would like to thanks to the reviewer for the time they have spent in reviewing our manuscript drafts R1 & R2 and for their wise comments to improve overall quality of the manuscript.
We have made all the corrections/changes suggested by the reviewers. All the authors have invested sufficient time in revising the manuscript and fix all the errors as suggested. All authors are agree for submission of revised version of the manuscript.

Looking forward to your response
With best and warm regards
